# Role of Different Abiotic Factors in Inducing Pre-Harvest Physiological Disorders in Radish (*Raphanus sativus*)

**DOI:** 10.3390/plants10102003

**Published:** 2021-09-24

**Authors:** Ayesha Manzoor, Muhammad Ajmal Bashir, Muhammad Saqib Naveed, Kaiser Latif Cheema, Mariateresa Cardarelli

**Affiliations:** 1Barani Agricultural Research Institute, Chakwal 48800, Pakistan; manzoorayesha12@yahoo.com (A.M.); saqibnaveedshah@gmail.com (M.S.N.); 2Department of Agriculture and Forest Sciences (DAFNE), University of Tuscia, 01100 Viterbo, Italy; muhammadajmal@unitus.it; 3Pulses Research Institute, Faisalabad 38000, Pakistan; klcheema@hotmail.com; 4CREA Research Centre for Vegetables and Ornamental Crops, 84098 Pontecagnano, Italy

**Keywords:** root, temperature, irrigation, hollowness, sowing, internal browning

## Abstract

Radish, one of the important root vegetables, is widely grown in the world due to its easy cultivation, short duration, growing habit, and adaptability to various growing conditions. However, it is still extremely difficult to produce good quality radish roots due to its vulnerability to different preharvest physiological disorders. Important physiological disorders that significantly reduce the yield and quality of radish are forking, pithiness/sponginess, cracking/splitting, hollowness, and internal browning. Different abiotic factors like moisture stress, temperature fluctuation, growing medium, nutrient imbalance, plant density and harvesting time cause a disturbance in the metabolic activities of root tissues that produce non-marketable roots. Therefore, this review provides a detail insight on the causes, physiology of these disorders, and the management practices to prevent them to produce commercial quality roots. This comprehensive knowledge will not only help the growers, but it will provide relative information for researchers as well to control these disorders through breeding innovations and biotechnological tools.

## 1. Introduction

Radish (*Raphanus sativus*), a member of *Brassicaceae* family, is one of the easy growing, fast maturing, and cold season herbaceous crops that have been cultivated for thousands of years. It is a rich source of starch, carotenoids, glucose, and other nutrients [1,2]. Radish storage roots are developed from two parts; the upper part consists of swollen hypocotyl, while the lower part is enlarged primary root tissues and lateral roots [3]. These two parts combine to form tuberous roots that are consumed raw as a salad with side dressings or cooked as vegetables [4]. Radish roots vary greatly in shape, size, color, and other external characteristics. Although it is a cold season crop, it can also be grown under high temperatures; however, the best quality root production is achieved at a 10–15.5 °C temperature [5]. Despite its adaptability to various climatic conditions and soil types, the successful production of radish is limited by non-marketable yield, which is associated with various physiological disorders that produce misshapen and deformed roots [6]. The best quality radish must be free from diseases, pests, and internal defects, have a crunchy internal texture, and freedom from bolting [7]. Therefore, the aim of this review is to summarize the effect of various abiotic factors on the initiation of physiological disorders and their management through altering growing conditions.

## 2. Physiological Disorders in Radish

Physiological disorders are malfunctions or dysfunctions of the physiological process of tissues [8]. They are described as a group of disorders that are caused by abiotic nonpathogenic factors such as environmental stresses (water stress, humidity, temperature, air pollutants), unfavorable soil conditions (poor soil drainage and extreme changes in soil pH), chemicals (pesticides, herbicides), inadequate concentration of growth regulators, nutrients excess or deficiency, etc. [9,10]. They are usually initiated before harvest, but appear after harvest, commonly during the storage period [11]. Physiological disorders also occur due to genetic susceptibility, enzymatic activities, over ripeness, aging, or due to cultural practices [12,13]. Usually, the symptoms of disorders look alike to those of diseases, but they can be prevented by altering environmental conditions; however, once they occur, they are irreversible [9]. Almost all vegetable crops are susceptible to various physiological/abiotic disorders affecting various plant organs, thus making them unfit for customer use [14].

As stated above, in vegetables, any type of abnormality in an important part of the plant that contributes towards quality and yield is named as physiological disorder [15]. Similarly, in radish, imbalance fertilizer application, inappropriate irrigation, temperature fluctuation, and various cultural operations under different environmental conditions produce pithed, hollow, cracked, forked, and malformed roots at different growth stages, which cause significant market loss (Figure 1 and Table 1). These disorders are visible only within roots and can be observed after harvesting or slicing.

### 2.1. Forking/Branching

It is a disorder in which the root changes into a fork-like structure [15]. Forking is an undesirable character that significantly deteriorates its quality and commercial value. It is described as a secondary root growth elongation that gives a fork-like appearance to the roots (Figure 2) [33]. Forking in root crops is associated with the use of undecomposed organic matter and the presence of high/excessive moisture content in the soil during the period of root development, and this disorder usually occurs in roots cultivated in heavy soils because of soil compactness [15,34]. Soils with poor physical properties such as having large lumps of soil and stones below the roots and shallow cultivation produced forked/branched roots [35]. Moreover, the use of old seeds having low vitality or affected by nematodes also developed forking disorder. Other factors such as damage to the meristem of the main root, curved hypocotyl after germination, and delayed harvesting have a role in inducing forking/branching [35,36].

In order to prevent this, well-decomposed manure and irrigation should be used at the right amount with the right quantity [15]. Varieties with round and short root shapes with rapid root growth must be planted in shallow arable or light soils, while varieties with long and large root shapes with poor ability of root elongation have to be planted in deep soils [33]. Furthermore, close spacing also (45 × 10 cm) leads to a higher % of forking due to increased competition for light, nutrients, and water among plants, which significantly affect plant growth, quality, and yield [17,37].

### 2.2. Pithiness

Pithiness is one of the main disorders of radish roots, it is also known as sponginess. It occurs when the accumulation of nutrients/assimilates lag behind the sudden increase of root volume because of abrupt growth (thickening), this inconsistency in growth results in the development of pores in roots [38]. It can be described as a rapid enlargement of root cells with insufficiently accumulated content of parenchyma cells [39]. Radish root quality is severely affected by sponginess or the development of internal cavities due to necrosis of parenchyma cells or breakage of xylem vessels because of rapid root elongation that increases root size [40]. Breakage of xylem vessels in response to strong elongation of root cells is considered as a first step in pithiness. Furthermore, cell size and distance increased with root growth, which leads to more pithiness [41]. Afterwards, the water content in root cells decreases and causes them to become sponge-like [35]. Sponginess in its final state causes totally hollow roots [39]. Incidence of sponginess occurs 7–8 days after the beginning of the elongation process [42]. Overall components such as rapid root growth, large cells, large distance in vascular tissue strands, and rupture of vascular tissues have resulted in pithiness disorders. Pith tissues develop from the largest cells, which are far from the vascular tissue strands and are located halfway between the center and periderm of the root [41]. Pithiness is also described as disintegration of the root heart, which occurs due to poor cell wall firmness and this is attributed due to calcium deficient soil [20].

Pithiness in radish is similar to the pith autolysis syndrome occurred in many dicotyledonous herbaceous plants. In this syndrome, the storage pith present in the lower stalk, petiole, and stem is autolysis by its own enzyme (cell wall degrading enzyme) causing a hollow stem [6]. Pithiness is the most important limiting factor that affects the postharvest life of radish, as it is in the final stage to develop hollowness in roots [38]. An approximately 100 mm^2^ in the longitudinal section of the root is considered as unacceptable. Pithy roots are characterized by textured white streaks, spots, or a network of dull white tissues that contrast with normal white tissue when observed through cross-section (Figure 3) [7].

It occurs due to delay in root harvesting [34]. Furthermore, bolting, over maturity [6,43] and excess of micronutrients (N, P, K) causes pore formation in roots. Therefore, to avoid pithiness incidence, maintain optimum soil moisture, harvest roots at the right time, avoid excess fertilization, and prevent injury to the roots during intercultural operations and harvesting [15]. As high nitrogen (N) and moisture content in growing media promotes pithiness. It is studied that organic manure such as rapeseed cake reduced pithiness incidence due to slow release of inorganic N that promotes stable root enlargement, whereas the use of bark compost increased the chances of pithiness in roots due to their high moisture content [39]. Wider spacing (45 × 20 cm) also promotes pithiness as it allows roots to utilize more resources which causes overgrowth and leads to pore formation [44]. Both low temperature and high soil EC (salt concentration) reduced the growth rate, which restricted the development of spongy tissues in the root [38]. In a study, high solar radiation (20 °C) with wider spacing (high inter-plant distance) causes increased production of dry matter that leads to pithiness development [6,41] and harvesting after 60 days of sowing may also cause this disorder [18].

Similarly, radish grown in the spring–summer season has high chances of pith tissue development compared to radishes grown in the autumn–winter season [7]. Among different growing substrates, sponginess was observed in sand more significantly than rockwool due to the rapid growth rate of the plant in sand [21]. Moreover, the reuse of growing substrate reduces sponginess in tissues due to an increase in EC, which significantly reduced growth [45]. The role of cultivation methods and hormones also initiates pithiness in roots, as foliar application of GA3 (40 mgL^−1^) on radish plants growing in furrows has a higher incidence of pithiness due to high plant growth rate [20].

### 2.3. Splitting/Cracking

Splitting is described as radical longitudinal fractures (Figure 4), which occur normally before harvest or observed during postharvest operations (1–2 days after harvest in storage). Its incidence can be as high as 30% and it can significantly reduce the marketable yield [46]; however, the commercial market tolerance for splitting can be less than 10% [47]. Root cracking/splitting during growth causes a significant reduction in yield because it delays flower initiation, pod formation, siliqua maturation with production of few seeds/pods [3]. Splitting disorder occurs in roots when the mechanical force exerted is greater than the ability of tissues to withstand it as some areas of tissue are more susceptible to splitting compared to others. Usually there are two types of splitting, one is “cellular debounding” in which cells remain intact but are pulled away from each other, whereas in the second type, “plasmoptysis”, cells burst/rupture. Both types of splitting depend upon the relative strength of intercellular bonds and cell wall integrity. Usually in radish splitting occurs due to plasmoptysis [47]. Splitting in winter radish is due to the difference in expansion rate between the periderm and internal tissues. As internal tissues expand more quickly than the periderm, putting pressure on these secondary tissues and causes them to split [48]. Factors such as external pressure, cell rupture or because of excessive tensile strength due to internal cell turgor led to hypocotyl splitting in radish [4].

There are many causes of radish root cracking, one of the most important is inadequate irrigation, which causes uneven growth between parenchyma cells in xylem vessels and periderm and phloem cells [3]. Moreover, irrigation frequency and quantity affect the water content of hypocotyl, which leads to the appearance of cracks on radish root during active secondary growth [46]. During growth, the availability of water and fluctuations in soil water potential led to the splitting disorder of radishes. As radish, that was irrigated once in three days has well developed hypocotyl and less cracking/splitting incidence compared to radishes which were irrigated daily, once in two, four, six or eight days [47]. Irrigation too frequently develops a high humid region in the root zone, which affects root growth due to inadequate oxygen diffusion in the root zone, but also due to wet soil, which causes parenchyma cells to expand quickly, but cells in the periderm and phloem do not expand accordingly that leads to root cracking, whereas too long irrigation interval induced water stress that restricts root growth and increase lignification of periderm [48]. High rate of cracking in radish roots (9.258%) occurred in irrigation treatment consisting of 100% available water capacity (AWC) depletion. Increase in root cracking is also linked with irregular cell growth and low transpiration rate. Since the translocation of calcium is linked to the transpiration stream, therefore, a reduction in transpiration causes calcium deficiency, which leads to the development of uncompact cells having fragile cell walls [6]. As the water content of the growing medium significantly influenced hypocotyl splitting at harvest, thus in another study, the dry period (no irrigation) during mid-growth resulted in fewer splits on hypocotyl compared to treatments that provide irrigation throughout the growth period [46]. Moreover, the use of perlite as a growing medium produced cracked roots due to inadequate water availability [31]. Along with the growing medium, the use of organic fertilizer (70 t ha^−1^) also developed split/cracked roots due to high nitrogen concentration [49]. In radish root development, 4 weeks after sowing is very critical, as during this period the root is expanding rapidly and any kind of irrigation or fertilizer stress during this period is important in producing cracked roots [24]. Likewise, irrigation and wide fluctuation in temperature during the later growth period leads to hypocotyl splitting [33] such as a high temperature around 35 °C accelerates lignin production around cells and results in the development of cracks outside radish roots [50].

Incidence of spitting is more in long rooted varieties, which are sparsely cultivated [33]. Boron deficiencies also appear to be one main factor in inducing hypocotyl splitting, as it is required for cell wall and membrane formation, therefore its deficiency will lead to plasmoptysis [47]. Furthermore, calcium deficient soils have a role in producing crack roots [50]. As cracks appear on roots during secondary growth (thickening), genes (*RsCDPK* and *RsANNAT*) involved in calcium transduction, binding, and ion transport increase in the expression level in cracking tolerant lines. Thus, these genes may influence root cracking by regulating the Ca2+ concentration signaling and help in cell proliferation and expansion. Thus, low concentration of calcium and low expression of the gene causes root cracking [3]. Therefore, roots growing in soil with high content of potassium (K) develop a high percentage of cracking of hypocotyl due to reduced absorption of calcium by the plants, as potassium activates a competitive inhabitation interaction with calcium leading to Ca2+ deficiency in plants [50].

Furthermore, splitting/cracking not only affects root quality, but also provides an entry to different pathogen (bacterial and fungal) invasions as shown in Figure 4 [51], which significantly reduce postharvest storage and shelf life [52].

### 2.4. Hollowness

Hollowness is defined as a lengthwise hollow cavity in the center of the root and usually appears in a summer grown radish crop (Figure 5) [28]. Hollowness occurs when intercellular air spaces fuse together at the center of the root (stele) near the pith during the first half of the growth period. Usually, these intercellular air spaces are filled with large spherical parenchymatous cells, but these cells fail to divide quickly to prevent the spaces to coalesce (combine/fuse) into a hollow zone under high soil temperature (35 °C) [1,53]. Moreover, the reduction in meristematic activity in xylem parenchyma cells located in the pith region causes retardation of inner cell formation [28]. High soil temperature also leads to lignification of cells facing the intercellular spaces in the center of the root and prevents rapid cell intrusion into these spaces and thus causes cells to coalesce to have developed a hollow cavity. It is observed that the inhibition of intrusion of rapid cells by lignification is the main cause of hollowness in radish roots [54] as lignified cells neither divide nor grow again [28]. After 21 days of sowing, intercellular spaces become macroscopic and the % of hollow spaces increased onward from the 28th day till harvest due to rapid root growth and cell thickening [55]. Hollowing disorder is severe in roots whose central region has the ability to expand, whereas, the root whose central part does not expand actively has a lower incidence of hollow cavity. Overall, hollowness initiation in the central region is affected by the obstruction of cell formation inside the intercellular spaces during early growth and rapid anatomical development (lignification) during root maturation. Favorable conditions for root growth such as early sowing in summer, high fertilizer application, and lower plant density are important in inducing hollowness. The more rapidly the roots grow, the higher chance of hollowness. However, unfavorable growing conditions such as late sowing, higher plant density, early defoliation, and low fertilizer rate, which prevent roots from growing well, result in lower or no chance of hollowness in roots [56]. Growth regulators also have a role in the development of hollow cavity in roots such as endogenous cytokinin has higher activity in roots clear from hollowness, whereas in roots with hollow cavity (HC), lower cytokinin activity is detected. Cytokinin has a role in promoting cell division and inhibiting lignin formation [57], whereas on the other hand, auxin accumulation in plant tissues inhibits cell division and accelerates lignification in parenchymatous cells [58]. Thus, hollowness incidence is affected by endogenous hormone activities that effect cell division and lignification. Therefore, HC in roots can be controlled by exogenous application of auxin and cytokinins [57].

Generally, various factors like reduction in cytokinin biosynthesis, high soil temperature, and loss of xylem parenchyma cells during the early to mid-growth stage, which causes breakdown of cells and formation of lysigenous intercellular spaces in the root and accumulation of lignin in them, leads to development of hollow cavity. Furthermore, when roots enlarge, hallow spaces remain because they cannot be filled with parenchyma cells [35]. In a study, a high soil temperature approximately (30.7 °C) during the 16–30th day after sowing causes a reduction in cytokinin activity in roots, which lowers the proliferation of xylem parenchyma cells in the intercellular space and leads to the development of hollow spaces in roots [59,60]. Similarly, [61] observed that early sown radish (July) has more incidence of hollowness compared to late sowing (August) due to exposure to high temperature (37 °C) during the middle of a growing season. It was also examined that the xylem vessel in both sowings was diverged into two sectors. However, the gap between the two sectors in early sown radish roots widened rapidly, causing the formation of a hollow cavity. In addition, high nitrogen application (300–450 mgL^−1^) causes hollowness in roots [40].

Hollowness in roots can be suppressed by spraying radish plants with high concentrations of alpha-naphthalene acetic acid (NAA), which severely reduced the root growth [1]. Moreover, spraying of radish plant leaves with 1-(2-chloro-4-pyridyl)-3-phenylurea (CPPU) at 10 mgL^−1^ and benzylaminopurine (BA) at >1 mgL^−1^ during the mid-growth stage significantly reduced the size of hollow cavities in root because of active meristematic activity of parenchyma cells that filled the intercellular spaces in pith with thin-walled cells and due to suppression of lignification in cells [54,57].

### 2.5. Internal Browning/Brown Heart/Akashin

Brown heart or internal browning (IB) is defined as the internal brown/reddish coloration in the central region of radish roots, which significantly reduced the commercial value of radish due to unpleasant appearance and bitter taste, and it is more common in summer season crops [31,62]. This disorder does not occur until the tap roots attain a vigorous growth stage, and subsequently brown substances start accumulating in tissues with the growth of tap roots [63]. The affected roots appear distorted, small and have a greyish appearance. Plants remain stunted due to reduction in growth, the leaves appear to have a variegated appearance with purple and yellow red blotches and roots along with showing brown coloration upon cutting also developed thick periderm [64,65,66].

The incidence of this disorder appears due to low boron concentration in soil. Boron requirement for radish is more than any other crop and its deficiency due to drought conditions or because of high temperature can cause internal browning [35]. In plants, boron has an important role in the proper functioning of cell wall function, as it helps in the formation of boric acid esters between pectin chains present in the cell wall, which helps in the maintenance of the cell wall (Figure 6). Therefore, variations in pectin concentration have a role in developing resistance to the brown heart by improving cell wall integrity [31]. Application of 15–20 kg/ha borax in soil and 0.1% B as foliar application is helpful in reducing this disorder [64].

Incidence of this disorder also correlated with the fluctuation in the activity of enzymes involved in the ascorbate-glutathione cycle and polyphenol biosynthesis in response to high temperature (Figure 6). According to this study, daily high temperature, usually above 21 °C, causes a reduction in ascorbate–glutathione cycle capacity to breakdown intercellular reactive oxygen species (generated due to heat stress) and increased activity of polyphenol oxidase, which leads to internal browning [32,67]. Time duration during which roots are exposed to high temperature is important in the incidence of IB as a 35 °C for maximum air temperature and 29 °C for maximum soil temperature is important in inducing brown heart disorder [30]. Reduced ability of the ascorbate–glutathione cycle to decompose H_2_O_2_ is responsible for the detoxification of H_2_O_2_ by polyphenol biosynthesis [68] as detoxification of H_2_O_2_ by PPO instead of glutathione reductase is pathway adaptive in plants in response to survive the toxicity of H_2_O_2_ [69]. Normally, the polyphenol oxidase is located in the plastid in an inactive form where it is bound to thylakoid membranes and phenols which serve as substrate for enzymatic browning are present inside the vacuoles. Thus, the enzymatic reaction leading to tissue browning does not occur in the cell at all. However, when surrounding membrane lipids are damaged by intercellular ROS, their activities are initiated in response to disruption in cellular integrity, both PPO and phenols leak from each cell organ into the cytoplasm and the enzymatic reaction is activated which leads to tissue browning [67,70]. IB sensitive radish roots showed high activity of polyphenol oxidase (PPO) and PAL (L-Phenylalanine ammonia-lyase) with reduction in glutathione reductase (GSHR) and dehydroascorbate reductase (DHAR) activities [70]. As moderately susceptible and susceptible cultivars have 1.2–3.8 times more polyphenol oxidase activity compared to resistant cultivars [32]. However, it was studied that the application of Sulphur in soil (30 gm^−2^) can increase the activities of ascorbate peroxidase (APX), GSHR and DHAR that inhibit enzyme activities involved in polyphenol biosynthesis thus increasing the decomposing capacity of ascorbate–glutathione cycle to scavenge H_2_O_2_ [71].

Incidence of internal browning is also linked with root anatomy as roots having small parenchyma cells have a low chance of brown heart disorder because small root cells contain a high amount of pectin polysaccharides that act as a barrier against ROS damage while roots have large cells, and active cell enlargement reduce pectin polysaccharide content that causes membrane damage by H_2_O_2_ and results in internal browning [62]. As the severity of this disorder in different varieties of radish largely depends on protopectin (water insoluble content, pectic substances) and the extent of cell enlargement. Cultivars or varieties with a small cell size (<135 µm) contain a high content of protopectin in roots and are resistant to internal browning as high concentrations of protopectin protect the cell membrane from degradation, whereas in moderately susceptible and susceptible cultivars, active cell enlargement (1.2–1.8 times larger cells than the resistant cultivar) reduced the concentration of protopectin due to membrane damage by ROS, which causes cells to collapse and increased activity of polyphenol oxidase that leads to enzymatic browning [32]. Moreover, exogenous application of GA3 during the mid-growth period also promotes small cell development, which suppresses internal browning in roots [62]. Therefore, a foliar spray of GA3 (100 ppm) significantly reduced the accumulation of brown substances due to the decrease in polyphenol oxidase activity and polyphenol contents in the root [72]. Furthermore, an inhibition of the regeneration system of ascorbic acid (AsA) causes disruption of the ascorbate glutathione system to detoxify H_2_O_2_, thus it was studied that in IB resistant cultivars, the level of AsA is 17–25 mg%, whereas in sensitive cultivars, the level reduced to 12 mg% [73].

As the susceptible cultivar has large parenchymal cells, higher respiration rate and has a high concentration of reducing sugars, i.e., sucrose is breakdown into glucose and fructose to maintain the cellular activities. Thus, having an elevated level of reducing sugars in root cells of susceptible cultivars also leads to Maillard reaction (a reaction between reducing sugars and amino acids) in cells in response to high temperature and oxidative stress. Maillard reaction in correspondence with enzymatic activities (PPO), results in internal browning in radish roots [67]. This it was obvious from different studies that incidence of this physiological disorder not only occurs due to different environmental and cultural stresses, but also affected by varietal differences [32,69].

## 3. Field Observations

During 2019–2021 (August–November), a radish germplasm evaluation trail under rainfed conditions at Barani Agricultural Research Institute, Chakwal, Pakistan was established to check the incidence of different physiological disorders. It was observed that all the above-mentioned disorders were appeared in different cultivars that were attributed to several abiotic disorders. Various cultivars were affected by only one particular disorder, whereas some of the cultivars were found to be susceptible to more than one physiological disorder, which caused a significant reduction in marketable value. The visual observations from that trial are mentioned in Table 2, in order to identify the occurrence of different disorders in different cultivars of radish.

### Strategies to Reduce Physiological Disorders

During field observation, it was noted that different abiotic factor such as soil type, high temperature, imbalanced fertilization, uneven irrigation, plant spacing, and harvesting time are the main causes behind these disorders. Therefore, different pre-harvest strategies or measures have to be taken to reduce or minimize the incidence of these disorders.

(a)Forking

For reducing forking, the plant radishes in light soils (sandy loam) with optimum spacing (10 × 75 cm) and adequate moisture during root development. Furthermore, harvesting roots after 40–60 days depends upon the cultivar type to prevent excessive secondary root elongation.

(b)Pithiness

As pithiness or pore development is related to rapid root development, thus it is important to avoid excess nitrogen (N) fertilization, low plant density, high soil moisture, and delay harvesting. Use a growing medium having high EC and prevent injury to roots during intercultural operations

(c)Splitting/cracking

The main cause of root splitting is the fluctuation in irrigation frequency, therefore irrigating radishes every 4–5 days during winter. Moreover, avoid sowing radishes in soil that cannot retain moisture (sandy soils). Additionally, remove weeding in radish fields every 15 days to minimize competition and retain moisture. According to this study, long rooted cultivars (purple neck, minowase, green neck) are more susceptible to splitting, therefore selecting small and round root radishes for sandy loam soils.

(d)Hollowing

Like pithiness, hollow space is also related to rapid root growth, and it occurs mostly due to temperature fluctuation during the primary root thickening stage. Therefore, sow radish cultivars (minowase, green neck, narrow leaf, black ball) in winter (Oct–Nov) during which the maximum and minimum temperature ranges from 28–10 °C.

(e)Internal Browning

To reduce IB incidence, plant radishes in spring to early summer season to avoid high temperatures during secondary growth. In addition, apply boric acid fertilizer in boron deficient soils.

## 4. Conclusions

Pre-harvest physiological disorder of radish is an abnormality of the root both externally and internally, that is, neither occurring by disease or by a mechanical agent, instead it is defined as a plant response towards different adverse growing conditions. Among different abiotic factors, high temperature is the main factor that develops various disorders in radish roots. Thus, it is important for radish growers to identify these disorders and alter different growing conditions such as sowing time, planting density, irrigation frequency, growing substrate, fertilizer rate, and proper selection of specific cultivars for specific conditions, etc., to mitigate these disorders.

## Figures and Tables

**Figure 1 plants-10-02003-f001:**
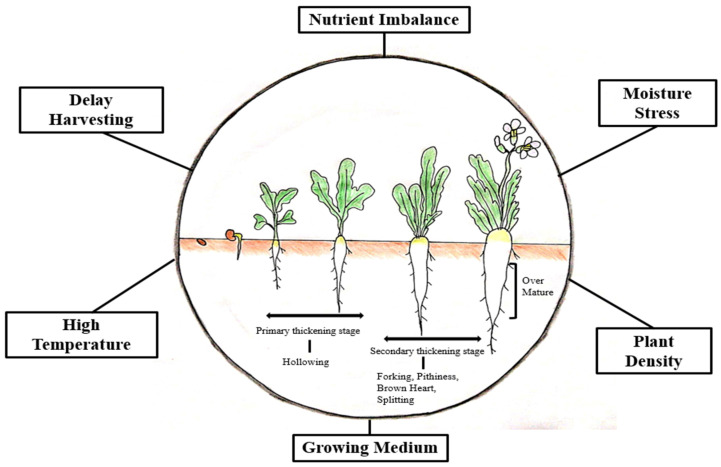
Abiotic factors induced physiological disorder in radish at different root development stages.

**Figure 2 plants-10-02003-f002:**
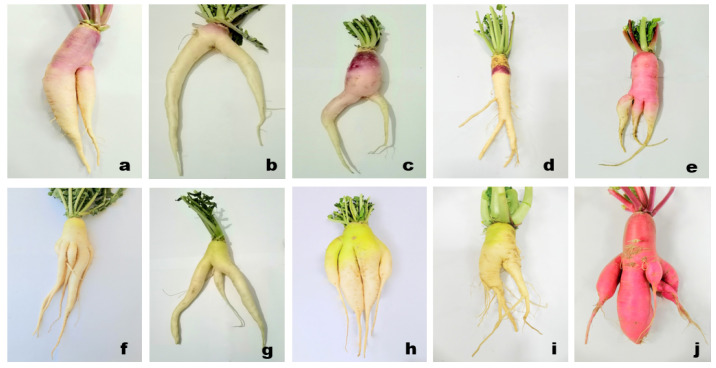
Different type of forking disorder in various radish cultivars: (**a**–**e**) (Purple neck), (**f**–**i**) (Green neck), (**j**) (Bari red).

**Figure 3 plants-10-02003-f003:**
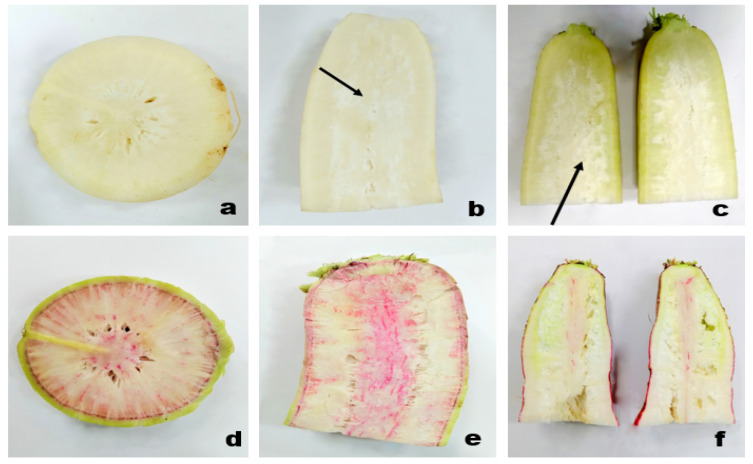
Pithiness disorder induction in different radish cultivars: cross section of pithed roots ((**a**): Minowase), moderate pithiness exhibiting dull white tissue in the longitudinal section of young roots represented by black arrow ((**b**): Bari white, (**c**): Green neck), over mature root that leads to pore development in cross section ((**d**): Lalpari), and longitudinal section ((**e**): Lalpari, (**f**): Bari Red).

**Figure 4 plants-10-02003-f004:**
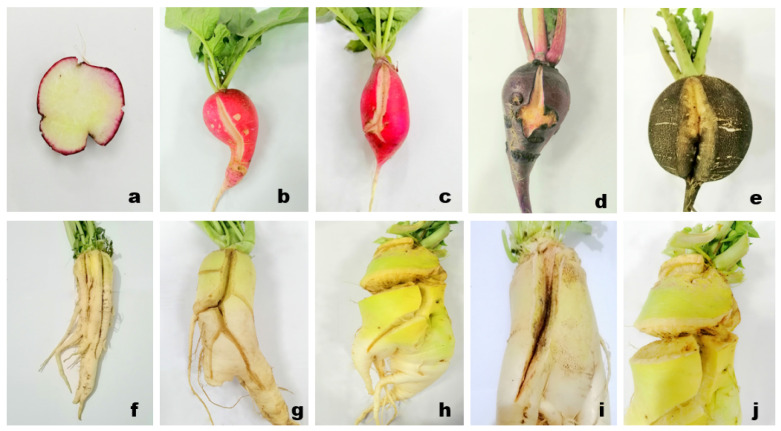
Cracking/splitting disorder in different radish cultivars: cross section of cracked roots ((**a**): Bari Red) moderate splitting: single longitudinal crack ((**b**,**c**): Belly red, (**d**,**e**): (Black ball)), severe splitting: 2–4 cracks on a single root ((**f**–**h**): Green neck), close view of cracked section showing fungal infection initiation: blackish growth ((**i**): Minowase, (**j**): Green neck).

**Figure 5 plants-10-02003-f005:**
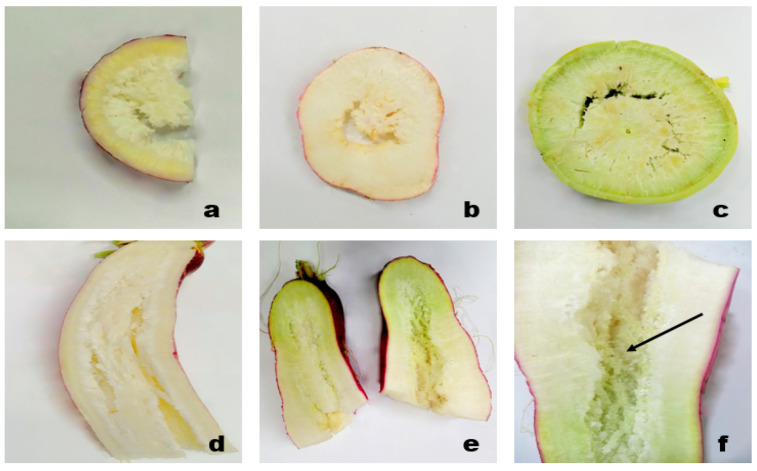
Hollow space formation in radish roots of different cultivars: cross section of hollowness in young roots ((**a**): Black ball, (**b**): Bari red) cross section of over mature hollow roots ((**c**): Green neck) longitudinal view of severe hollowness in young roots ((**d**,**e**): Lalpari) close view of hollow space: formation of distorted cells represented by black arrow ((**f**): Lalpari).

**Figure 6 plants-10-02003-f006:**
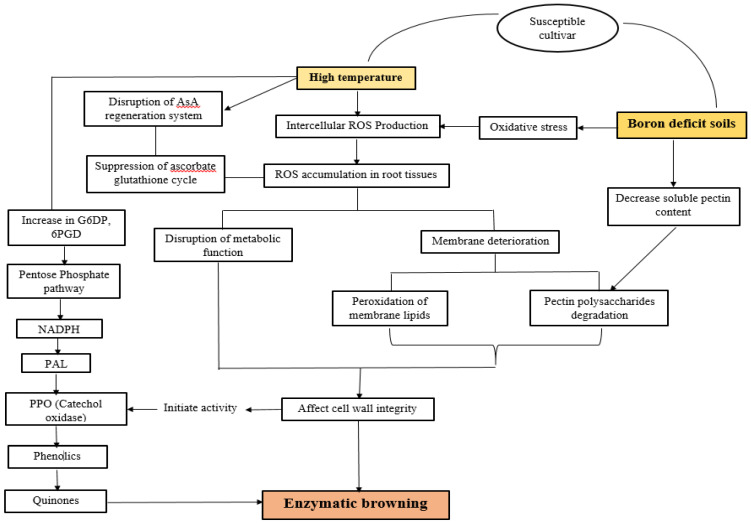
Brown heart/internal browning disorder in susceptible cultivars of radish is due to oxidation (enzymatic) of phenolic compounds and it occurs mainly by two important factors. (1) Low boron concentration in soil. (2) High temperature. Both factors induce the generation of reactive oxygen species (ROS) in cells that degrade pectin polysaccharides in the cell wall, which reduce cell integrity. High temperature causes a reduction in the ascorbic acid regeneration system in the ascorbate glutathione cycle that limits its capacity to detoxify ROS. Furthermore, high temperature also leads to increased activity of glucose-6-phosphate dehydrogenase (G6PD) and phosphogluconate dehydrogenase (6PGD) enzymes that initiate the pentose phosphate pathway (PPP) to provide energy in the form of NADPH to breakdown ROS by polyphenol oxidation (increasing activity of PAL and PPO).

**Table 1 plants-10-02003-t001:** Effect of different abiotic factors on incidence of physiological disorders in radish and their remediation.

Disorder	Abiotic Factor	Reason	Control	References
Forking	Undecomposed organic matter in soil	Excessive root elongation	Use well decomposed organic matter	[16]
	High plant density	Competition between moisture, light and nutrient increase	Maintain optimum plant density	[17]
Pithiness	Delay harvesting	Development of larger cells leads to pithiness	Harvest radishes at right time	[18]
	Plant spacing	Double row spacing (45 × 25)/wider spacing causes root enlargement	Plant radish at recommended plant and row spacing	[19]
	Cultivation method	Furrow cultivation leads to rapid root growth	Cultivate radish on flat beds	[20]
	Growing substrate	Sand increase growth rate	Use well balanced growing medium	[21]
Splitting/cracking	High organic fertilization (nitrogen)	Cause parenchyma cells to expand at higher rate	Provide optimum nitrogen doses to avoid excessive cell growth	[22]
	High water content of growing medium	Rapid cell enlargement during secondary thickening	Maintain optimum level of growing media water content	[23]
	Irrigation irregularity			[24,25]
	Heavy metal toxicity in acidic soils	Aluminum toxicity leads to disturbance in cell anatomy	Improve soil pH	[26]
Hollowness	Growing medium (coir dust)Over maturity	Increase expansion of parenchyma cells	Cultivate radish on well balanced growing mediumHarvest roots at right time	[27]
	Varietal effect	Hollowness susceptible cultivars has rapid rate of cell enlargement	Plant cultivars tolerant to hollowness disorder	[25]
	Nutrient imbalance	High N causes cell expansion	Provide optimum nitrogen doses to avoid excessive cell growth	[1]
	High soil temperature	Increase cell lignification that promote hollowness	Plant radish during low temperature	[28,29]
Internal browning	High air and soil temp	Increase enzymatic activity	Controlled temperature during later growth period	[30]
	Nutrient imbalance	Boron deficiency affect cell wall firmness	Provide micronutrient at optimum rate	[31]
	Genetic effect	Resistant cultivars had high pectin content	Plant cultivars tolerant to brown heart	[32]

**Table 2 plants-10-02003-t002:** Abiotic factors induced different physiological disorders in various cultivars of radish.

Disorder	Cause/Abiotic Factor	Cultivar	Pictorial View
Forking	Delay harvesting, Soil with poor physical properties (hard lumps), Close spacing (R-R = 5 cm, P-P = 75 cm)	Purple neck, Green neck, Minowase, BARI Red	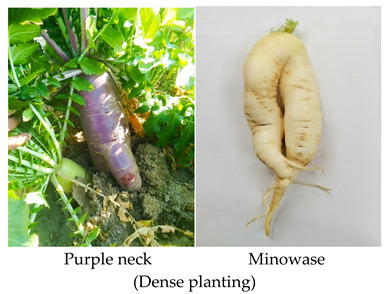
Pithiness	Over maturity, early sowing (August-October), High irradiance (32.6 °C), sandy loam soil which promotes rapid growth, low soil EC (0.8 ds/m)	Lalpari, Narrow leaf, BARI Red,	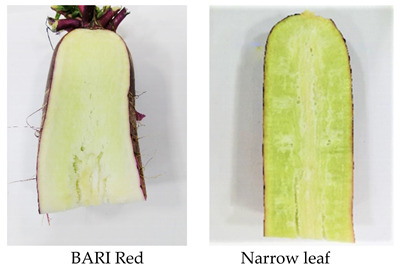
Cracking/Splitting	Heavy rainfall after a long dry period (40.4 mm), high temperature during root growth (28 °C) in late sown plot, over-maturity	Belle red, Narrow leaf, Green neck, Minowase, Black ball, BARI Red	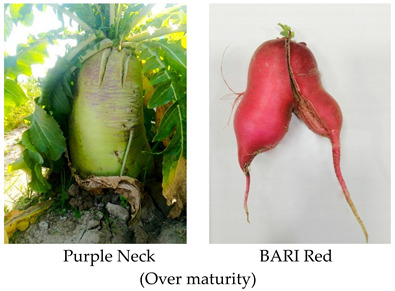
Hollowness	High soil temperature (36.4 °C) in early sown plot, over maturity	Narrow leaf, Purple neck	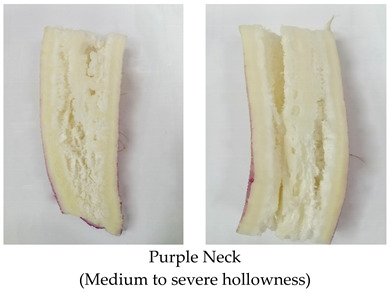
Internal browning/brown heart	High soil temperature during primary thickening growth stage (30–36.4 °C), boron deficient soil (0.35 mg kg^−1^)	Minowase	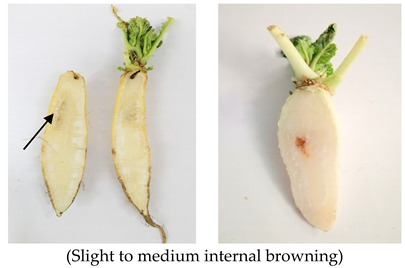

## Data Availability

Not applicable.

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
