# Peer review of "Role of Different Abiotic Factors in Inducing Pre-Harvest Physiological Disorders in Radish (*Raphanus sativus*)"

_plants, 2021, doi:10.3390/plants10102003_

Round 1
Reviewer 1 Report
Tab 1: References are missing
The figures are of poor quality (background shadows) and a scale is missing, which belongs to an adequate describtion of plant material. Also the name of the cultivars is missing and should be mentioned not only in some figures.
Research Study: The research study isn't well described and from the "result" it isn't clear, which influence the date of sawing has on the various disorders....no description of the plant material, how many replicates, which soil/ environmental conditions etc.
Is this research study published somewhere? In this case the reference should be mentioned.
One severe mistake is that the authors mention plant genotype as an abiotic factor like soil, temperature etc. (Abstract and Line 413/414) This indicates an ignorance of the physiological/ genetic facts.
Conclusion: climate change is mentioned as a factor influencing the disorder of the roots, but the authors didn't investigate "climate change" and it's influence on root development.
Author Response
Dear Professor. Thank you for taking time out of your busy schedule to review our manuscript, and providing us many valuable comments and suggestions.
We have revised the manuscript according to your comments.
Hoping for your positive reply, all the best
Reviewer 2 Report
This review article provides the timely and important information on the physiological disorders in radish, which was never published in any scientific journal recently. The article is well organized and contains many appropriate references for the subject. The information provided by the manuscript is helpful for the readers of Plants, especially for those working on radish and other root vegetables. I provided major comments and several minor suggestions below and hope these can help to improve the manuscript.
Major comments
- Lines 45-79 are general information on the physiological disorders and not directly related to the object of this article. These lines should be removed from the manuscript and the definition and types of physiological disorders, in two or three sentences, can be added between lines 81 and 82.
- Table 1 should be removed for the same reasons above.
- Lines 397-407 and Table 3 should be removed since there is no detailed information on the experimental designs, environmental controls for high irradiance and soil temperature, and the number of plants used. Besides, the normal plants from normal conditions did not provided. Otherwise, the author should provide the information mentioned above.
Minor suggestions
- Sowing date cannot be generalized since different regions have different radish growing seasons and sowing dates. Therefore, sentences containing ‘sowing’ and ‘sowing date’, for an example, lines 121 and 122, should be changed.
- Some grammatical errors can be seen in the manuscript although I am not a native English speaker.
Author Response
Dear Professor. Thank you for taking time out of your busy schedule to review our manuscript, and providing us many valuable comments and suggestions. We have revised the manuscript according to your comments
Hoping for your positive reply
all the best

Reviewer 3 Report
This is a really nice review on the "Role of different abiotic factors in inducing pre-harvest physiological disorders in radish (Raphanus sativus)". I enjoyed reading it. The manuscript is well supported by the literature cited and the figures are nice and clear. I only suggest a revision of the English language as a minor revision.
Author Response
Dear Professor. Thank you for taking time out of your busy schedule to review our manuscript, and providing us many valuable comments and suggestions. We have revised the manuscript according to your comments
all the best

Round 2
Reviewer 1 Report
The manuscript is improved, but nevertheless there are some small improvements to be made:
Fig. 5: add explanation for black arrow into the legend
chapter 3. field observation: please check sentence structure; to me there seems to be something wrong
Author Response
Dear Professor. Thank you for taking time out of your busy schedule to review our manuscript, and providing us some more valuable comments and suggestions. We have revised the manuscript according to your comments. In addition, a point-by-point response to your comments that enhanced the quality of our manuscript are mentioned

Reviewer 2 Report
The revised manuscript is now acceptable to the journal, Plants.
Author Response
Dear Professor. Thank you for taking time out of your busy schedule to review our manuscript, and accepting it suitable for its possible publication. Your comments helped a lot in improving this manuscript according to standards of scientific community. Moreover, the English language and grammar has also been checked again throughout the manuscript.